# Barrier Effect in a Medium-Sized Brazilian City: An Exploratory Analysis Using Decision Trees and Random Forests

**Mylena Cristine Rodrigues de Jesus *** and **Antônio Nélson Rodrigues da Silva**

Department of Transportation Engineering, São Carlos School of Engineering of the University of Sao Paulo, São Carlos 13566-590, Brazil; anelson@sc.usp.br
* Correspondence: mylenacrj@alumni.usp.br

**Abstract:** This study aims to examine if an urban road with intense motorized traffic in a medium-sized Brazilian city constitutes a barrier for walking trips. A questionnaire was conducted with 103 individuals in an area up to 800 m from the road selected for the study to obtain information about personal characteristics (age, income, etc.), social interactions in the neighborhood, and travel and mobility characteristics. We used the dataset to explore the potential of Decision Tree and Random Forest classification models to predict the users' perception of the barrier effect, which was characterized by the dependent variables speed and volume (of motorized traffic). For 36.9% and 47.6% of respondents, traffic speed and traffic volume, respectively, represent a barrier to walking. The results also show that the following variables considerably affect the perception of the barrier effect of the respondents: distance from their residence to the studied road, time living at the address and in the study area, social connections in the neighborhood, and the street reported as the busiest one in the neighborhood. Identifying the variables with the largest influence on the perception of the barrier effect may be very useful for planning and policy initiatives.

**Keywords:** barrier effect; community severance; non-motorized transport; pedestrians; active modes





## 1. Introduction

Joint actions carried out by various stakeholders in the transport system (pedestrians, cyclists, motor vehicles and land use activities) cause numerous conflicts, the effects of which are unevenly intensified for those involved. One of the most constant conflicts in the transport system is between non-motorized modes and motor vehicle traffic [1]. In this context, the barrier effect occurs when a transport infrastructure acts as an obstacle to local pedestrian and cyclist movements. This effect occurs more intensely in the case of crossing roads, but it is also observed for displacements along the road. In the case of urban streets and roads, more specifically, the speed and volume of motor vehicles act as barriers that compromise displacement, access to goods and services and the well-being of the population [2].

This type of situation is common in large cities, where the fleet of motorized vehicles is significant, especially the portion corresponding to individual motorized vehicles. This is not a frequent situation, on the other hand, in small cities. However, in this study we aim to understand medium-sized cities: is it possible to speak of a barrier effect in this context?

Due to this, in recent years, some researchers have sought to understand the effects of transport infrastructure on individuals' daily commutes [2–8]. Some of the results found by these authors show, for example, that the proportion of people with compromised well-being is higher among respondents who live closer to a busy road [2]. Furthermore, respondents who perceive traffic volume as heavy and traffic speed as high are more likely to report that traffic conditions represent a barrier to their walking trips, leading them to avoid walking in this area [9].

In order to contribute to the literature in this area, the main objective of this research was to identify whether signs of barrier effects for pedestrians can be observed, even in a

medium-sized city. To guide the study development, which focuses on analyzing an urban road selected for presenting a high traffic volume of motor vehicles, the following research questions (RQ) were considered:

RQ1. Is it possible to say that, in some respects, the selected road acts as a barrier to the study area?

RQ2. If so, what are the variables that influence the pedestrians' perception of the barrier effect?

RQ3. Is it possible to relate the pedestrians' perception of the barrier effect and their individual characteristics, social interactions, and travel and mobility characteristics?

To address these questions, the present study presents an exploratory analysis of the data obtained with a questionnaire applied to residents and regulars. The results that will be described are essentially based on the interviewees' opinions about the speed/volume of traffic in the study area, in addition to considering the individual characteristics of the respondents and the difficulties they face as pedestrians. The methodology is based on a descriptive and quantitative analysis of the data, using non-parametric Decision Tree (AD) and Random Forest (AF) techniques to predict the respondents' perception of the barrier effect, to identify the most relevant one in this context and to identify patterns among the perception of the users interviewed.

## 2. Literature Review

The community separation effect, or barrier effect, has been studied since the 1960s [10]; however, in Brazil, research in this area seems to have started only around the 2000s [11,12]. During these years, researchers have presented some aspects of defining the barrier effect, such as the consequence of motor vehicle traffic on pedestrian movement and on individuals' interactions [13,14], the relationship between traffic conditions and pedestrian behavior [15], problems caused by pedestrian movement as a result of static barriers [3], restrictions of pedestrian movement from one side of a road to the other, caused by traffic and road infrastructure [11,12,16], and local barriers to travel by non-motorized modes [4,6].

However, the most recent studies have defined that the barrier effect, caused by a transport infrastructure and/or by motorized traffic, results in a negative impact on the pedestrian behavior, well-being and mobility of individuals, preventing access to necessary goods, services and networks for a healthy life [2,10,17]. Some authors also show that the barrier effect is due to the speed at which cities are undergoing urbanization, which is responsible for their fragmentation, with neighborhoods divided by a road transport infrastructure and where motorized traffic dominates to the detriment of pedestrians [18].

In this context, some researchers have also identified the possible consequences imposed by this effect on the local community, making studies on this topic relevant to the Sustainable Development Goals (SDGs) proposed by the United Nations [19]. This is the case, for example, of SDG 11, which focuses on sustainable cities and communities and encourages actions to provide access to safe, accessible and sustainable transport systems for all, improving road safety, with special attention to the needs of people in vulnerable situations [20]. In this context, studies show that residents of streets with high motor traffic have fewer social interactions with their neighbors when compared to people living on streets with less traffic [13,21]; the most affected individuals are children and the elderly, followed by people with impaired mobility and/or adults who need to accompany them [12,14,17]. Furthermore, the authors of this previous research also show that the area considered as their neighborhood by people who live on busy streets is smaller than that of residents of streets with light traffic; also, the level of the barrier effect varies markedly according to the age group of pedestrians [15].

Other studies also address the influence of traffic volume and speed on displacements and pedestrian perception, such as: increasing the duration of walking and cycling trips, as people need to deviate from the shortest paths to cross the road in places considered safer [17]; influencing the balance between mobility and pedestrian safety, as well as pedestrian behavior at the crossing and their perceptions of the street [15]; constituting

barriers to mobility and accessibility, mainly in conjunction with the lack of crossing points and crossings with insufficient time for pedestrians to cross [22]; influencing the individual's perception of the barrier effect, as those who perceived the volume of traffic as heavy and the speed of traffic as high are more likely to report that traffic conditions represent a barrier to their journeys on foot, leading to them to avoid walking in this area [9].

In addition, van Eldijk et al. [23] pointed out, in a literature review, some direct indicators of the barrier effect, mapped by the studies developed so far, as shown in Table 1. Based on this, in the literature studied about the barrier effect caused by transport infrastructure and motor vehicle traffic, there was a lack of studies that consider this scenario in medium-sized cities, especially in developing countries. Moreover, many analyses in this context were performed using aggregated data, and it was not possible to sense the individuals' perception as accurately as in a survey based on disaggregated data. Therefore, the purpose of the present study was to analyze evidence that a road with high traffic of motor vehicles acts as a barrier to pedestrian movement in the study area, using disaggregated data. Furthermore, the aim was also to identify the most influential variables and possible patterns between the individuals' perceptions so as to continue the research, fill the gaps in studies already carried out and contribute to the evolution of studies on the barrier effect in medium-sized cities.

**Table 1.** Survey of direct indicators of the barrier effect, according to van Eldijk et al. [23].

| Effect | Indicator |
| --- | --- |
| Crossing effort (static characteristics) | Location of the transport infrastructure, width of the road, number of lanes, obstacles along the infrastructure, width of the median, visual conditions for crossing, level difference. |
| Crossing effort (dynamic characteristics) | Speed, volume, vehicles composition, traffic direction, parked vehicles, risk of traffic accidents when crossing, etc. |
| Crossing effort (facilities) | Distance to the crossing point, delay in the traffic light turning green, effort required to use the crossing facilities, protection against weather conditions at the crossing facilities. |
| Fear of crime | Social surveillance, escape options, visual conditions. |
| Travel effort | Distance and distribution of means of crossing, number of barriers along the routes, number of disconnected streets, etc. |
| Accessibility | Choice and possibility of substitution of destinations, accessibility to employment, separation degree, potential for loss of interaction between the population, land use connectivity, etc. |

In order to compare the results with those of other cities, the questionnaire and methodology used in this study were based on the study carried out by Mindell et al. [2] in the survey "Health and neighborhood mobility survey" [24], and the importance of simultaneously evaluating individuals' perception about the volume and speed of traffic and their influences on walking and well-being, highlighted by Anciaes et al. [9].

## 3. Materials and Methods

The methodology was applied to a medium-sized city, which, according to Brazilian standards, should have a population between 100 and 500 thousand inhabitants [25]. The study area was around Rua Miguel Petroni, a busy street located in the city of São Carlos, in

the state of São Paulo. This study began by distributing a questionnaire drawn up based on a set of tools proposed by Mindell et al. [2] to measure the degree of community separation due to busy streets and their impacts. The questionnaire has 37 questions and was designed to allow various analyses. The questions address issues related to the characterization of the interviewee (age, academic background, income, address, health status, among others); social interactions (for example: contact with neighbors, integration into the study area, trustworthiness of people, fear of walking alone at night, cleanliness of the study area); travel and mobility (walking difficulties, disabilities, number of walking trips and duration thereof, existence of a crosswalk, whether the speed and volume of traffic represent barriers to reaching a destination on foot, characterization of the street where the person lives, the neighborhood or street that he/she considers the busiest and other aspects); and, finally, an open space for the respondent to express his/her opinion (in this part, the answers were audio recorded). A summary of the independent variables considered is presented in Table A1.

Data were collected in field research, using electronic devices (cell phones, tablets, among other devices) through an interface created by the Open Data Kit (ODK) application, which allows data collection to be carried out offline and can extract tabulated data in electronic spreadsheets. In addition, the questionnaires through interviews were approved by the Research Ethics Committee for Human Beings at the Federal University of São Carlos, under Report No. 4,043,848, of 2020. The complete questionnaire can be seen in reference [26].

The distance considered walkable in a neighborhood may vary according to a pedestrian's health conditions, the infrastructure offered, among other factors [27,28]. For this study, a distance of up to 800 m was adopted, considered reasonable for walking in the study area, as well as in the research conducted by Mindell et al. [2]. Within this threshold we expected to find in the study area people likely to walk on Rua Miguel Petroni. As a matter of fact, only people who knew and had already walked down that street answered the questionnaire. Thus, residents, workers and regulars, who voluntarily agreed to participate in the research, completed a questionnaire in March 2020 in an area of up to 800 m from the road of interest. The questionnaire took approximately 20 min to complete. The interviews, carried out during business hours (from 8 am to 6 pm) from Monday to Friday, involved males and females aged 18 years or over.

### 3.1. Barrier Effect Analysis

According to the neighborhood health and mobility survey conducted by Mindell et al. [2], almost half of the participants reported that their ability to walk was at least occasionally affected by the speed or volume of traffic on the streets around their home. Therefore, these were the study variables in this case study, although other variables that influence the barrier effect are known. Therefore, focusing on the first research question, respondents were asked whether traffic speed and traffic volume constitute a barrier that prevents or makes it difficult for them to reach their destinations on foot (Q25_Speed and Q25_Volume). The alternatives for these questions were "never", "rarely", "sometimes", "usually" or "always". To facilitate data analysis, especially in cases where the number of observations is not very large, those alternatives can be grouped into two classes: those that do not consider (never and rarely) and those that do consider (sometimes, generally and always) traffic speed or traffic volume as a barrier to pedestrian movement. The output results in this case are binary variables of two classes, where YES (or Class 1) corresponds to speed or volume as a hindrance to walking and NO (or Class 0) corresponds to negation. The results of this analysis provided answers to RQ1.

### 3.2. Analysis of Respondents' Perception of the Barrier Effect

In order to answer the second and third research questions, the variables traffic speed and volume in this study are considered as dependent variables to apply the Decision Tree and Random Forest techniques. To this end, it is assumed that the sample obtained is

randomly divided into two groups, a part for training (80% of records—83 interviewees) and a part for model validation (20% of records—20 interviewees) and that the techniques are applied to four groups of different variables for each of the dependent variables. Thus, in principle, both speed and volume constitute, separately, dependent variables for analyses involving three groups of variables: (i) characterization of the interviewees, (ii) social interactions and (iii) travel and mobility. Finally, for each dependent variable, the variables highlighted as the most relevant are selected in the three groups previously analyzed and then, a new analysis (iv) is carried out based on the highlighted independent variables.

Regarding the analysis techniques used, the Decision Tree method sought, in this research, to classify the data collected in terms of the two characterizing variables of the study: traffic speed and volume. This classification was performed through the other variables in the questionnaire, which are believed to influence pedestrians' perception of the speed and volume of motorized traffic. Based on this, the variables that stood out in terms of the prediction and results presented by the tree were also observed.

The Decision Tree is a non-parametric method in which the observations are arranged in a tree-like format, in which the leaves characterize results and the branches the conditions based on the inputs to the model. In this case, the CART algorithm was used, and the leaves represent the number of people, or percentage of the sample, which indicates the speed or traffic volume as a barrier (Class 1) or not (Class 0), while the branches represent the independent variables that were important to reach this decision. Different Decision Tree structures were tested, varying the criteria used to prepare them, aiming to observe the variations in their performance parameters and, finally, choosing, for analysis, those that presented metrics (accuracy, precision, sensitivity, specificity, among others) with better results.

The Random Forest technique is a form of classification that consists of a set of independent Decision Trees, in which each resulting tree composes a vector and then computes a unit vote for the most popular entry class [29]. The generated set of votes is less sensitive to outliers than lone trees, in addition to improving the robustness of the results [30]. On the other hand, the technique loses in the precision of the interpretation, as it is not possible to analyze all the generated trees, but rather a set of important variables based on them. Based on its robust set of Decision Trees, this technique aimed to assess the importance of the variables and those that most contributed to the model's prediction. In the end, it results in a graph of important variables showing, within all the trees analyzed, which variables stood out the most.

The Decision Trees and Random Forests necessary for data analysis were generated using software R (R for Windows 4.0.5, Vienna, Austria) In each tree, based on the division criteria, the observations were divided into binary nodes until the best performance of the model is reached, that is, low impurity of the generated nodes. Impure nodes, unlike homogeneous nodes, do not have all the observations in the same class [31]. Considering this, the impurity measure used in this study is the standard Gini measure, in which the optimal value occurs when all cases of a node fall into a single category (i.e., YES, respondents consider traffic speed or traffic volume a barrier, or NOT, do not consider, for example).

As a result of both techniques, the independent variables that most influenced pedestrians' perceptions of the barrier effect were obtained. That is, to respond to RQ2, the importance of independent variables was observed both in the Decision Tree model and in the Random Forest model.

On the other hand, the classification resulting from the Decision Tree helped to address RQ3, presenting some characteristics that lead the interviewees to opt for each of the classes of the dependent variable and the existing patterns in the collected sample.

## 4. Results

The methodology presented above was used in a case study in the city of São Carlos, in the state of São Paulo, Brazil. The city has an estimated population of 254,484 people

(estimate for 2021 from the 2010 census). In addition, it has an urban territorial area of approximately 80 km² [32,33]. The city in question was chosen for this analysis because it is a medium-sized city (by Brazilian standards) and, due to the development and increase in vehicle traffic, it presents challenges for displacements by active modes in certain neighborhoods.

The specific area of the municipality selected for the study comprises a stretch of about 1.5 km of Rua Miguel Petroni (Figure 1—in Portuguese, Rua means Street), an urban road with intense traffic of motor vehicles and 4.9 km in total length. That part of the street was selected, similarly to what was done in other studies [2,5,6,9], because of previous reports of the population describing difficulties for crossing and dangerous situations for displacements by active modes. Based on the city's road system, the section was selected because it has a traffic volume varying between medium and high (600 and 1000 vehicles per hour, according to data provided by the Municipality of São Carlos) and poses a challenge for the pedestrian and cyclist movement, especially at peak times. Rua Miguel Petroni, which is characterized as a main street in that part of the city, is a one-way road, mostly with two lanes for circulation and a single parking lane on the left. In addition, according to the classification of the National Department of Transport Infrastructure [34], it is a secondary arterial road, with a maximum permitted speed varying between 50 and 60 km/h.

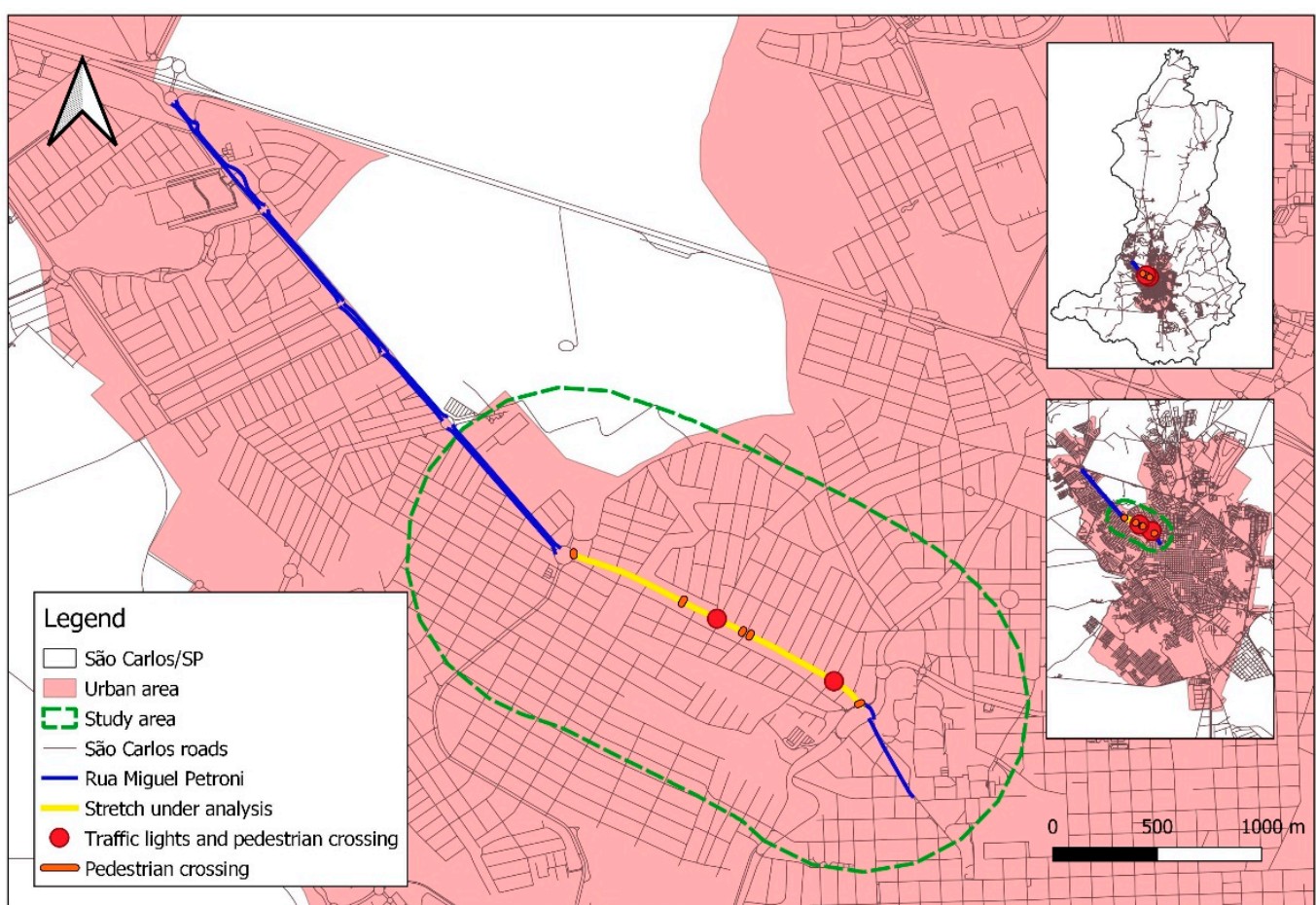

**Figure 1.** Location of Rua Miguel Petroni (São Carlos, SP) and the study area.

The buffer delimiting the study area was generated using GIS software (QGis 3.16.11 version, designed by the Open Source Geospatial Foundation—OSGeo, Beaverton, OR, USA) by considering a Euclidian distance. The 2010 census data [35] allowed us to estimate that about 15 thousand inhabitants live within 800 m from the selected section of the study

road (as shown in Figure 1), a distance adopted because it is considered reasonable for local travel, including walking. At first, to ensure that the sample collected had statistical representation, considering a confidence level of 95% and a sample error equal to 0.05, we originally planned to collect data from 385 people. However, data collection was interrupted due to the COVID-19 pandemic and, in order to continue the studies, it was decided to increase the sampling error interval to 0.10, with the same level of confidence. In this case, the minimum sample size would be 96. The sample collected for this research, however, has a non-probabilistic profile, because to reach the number of proposed questionnaires and due to limitations in the continuity of data collection, no strategy was adopted to select the homes and commercial establishments visited. Moreover, as pedestrians in circulation and some residents and workers in the study area were interviewed, it can be treated as a convenience sample.

A total of 103 respondents were interviewed, of which 66% were female, predominantly aged between 30 and 59 years, most of whom had completed high school, and about 86.4% considered their health status to be excellent or good. In addition, the data collected regarding housing allowed us to observe that the percentage of people who live in their own or rented/other property is approximately equivalent (50% each) and that 78.6% live at a distance of less than 400 m from Rua Miguel Petroni. It is also noted that 68% of respondents have lived in their respective addresses for more than 12 months (1 year). Some assumptions can be made about those interviewees who have lived at their current address for a long time: they know the study area better, as well as the positive and negative points; and have a better understanding of the neighborhood. On the other hand, they may have become so familiar with the negative points that they may no longer even notice them.

*4.1. Traffic Speed as a Dependent Variable*

The questionnaire allowed us to identify the respondents' percentage who believe that the traffic speed is an obstacle to their journeys on foot. In Class 1, accounting for 36.9%, are those who thought that speed is an obstacle to walking. Class 0, with 63.1%, corresponds to denial. Then, the dependent variable was analyzed using Decision Tree and Random Forest techniques in order to classify the observations regarding traffic speed, identifying which characteristics eventually lead the interviewees to choose each of the classes of the dependent variable and identifying the independent variables that most influence the respondents' perception of it.

From the variables that stood out as relevant in the analyses involving traffic speed as a dependent variable and the characterization of the interviewees (i), the characteristics of social interactions (ii) and of travel and mobility (iii) as independent variables, the fourth analysis was made. In this case, the variables selected as independent were: group i—distance from the residence to Rua Miguel Petroni (Q5_Distance), time that the interviewee has lived in the study area (Q7_TimeRegion), time that the interviewee has lived at the current address (Q6_TimeAddress); group ii—number of neighbors known on the same side and on the other side of the street he/she lives on (Q11_NeighborsSameSide and Q12_NeighborsOtherSide); and group iii—which is the busiest street close to where the person lives (Q31_BusiestStreet) and if the person avoids walking on that street because it is very busy (Q33_AvoidCrossing).

Among the Decision Trees generated, by alternating the composition criteria, the tree chosen for analysis (minsplit = 8, minbucket = 4, maxdepth = 5) was able to correctly predict 86.75% and 60.0% of the results of training and validation samples, respectively (Table 2). The tree in question, created considering the traffic speed and the most relevant variables, classifies the sample into 12 leaf nodes (Figure 2). Continuing the process, for the same configuration of dependent and independent variables, the Random Forest analysis was conducted. From the set of variables importance graphs, generated by both techniques, it can be seen that the distance from the interviewee's residence to the study route (Q5_Distance), highlighted with a dashed line in the importance graphs, is the

variable that stands out as important for the prediction of both models. The performance parameters of the models are shown in Table 2.

**Table 2.** Predictive performance of Decision Tree and Random Forest models (Traffic Speed and the most relevant variables) (n = 83).

| Performance Measures | Decision Trees | | Random Forests | |
|---|---|---|---|---|
| | Training | Validation | Training | Validation |
| True positive | 46 | 8 | 52 | 9 |
| False positive | 5 | 3 | 0 | 5 |
| False negative | 6 | 5 | 0 | 4 |
| True Negative | 26 | 4 | 31 | 2 |
| Accuracy (%) | 86.75 | 60.00 | 100.00 | 55.00 |
| Sensitivity (%) | 88.46 | 61.54 | 100.00 | 69.23 |
| Specificity (%) | 83.87 | 57.14 | 100.00 | 28.57 |
| Prevalence (%) | 62.65 | 65.00 | 62.65 | 65.00 |
| Positive Predictive Value (%) | 90.20 | 72.73 | 100.00 | 64.29 |
| Negative Predictive Value (%) | 81.25 | 44.44 | 100.00 | 33.33 |
| False Discovery Rate (%) | 9.80 | 27.27 | 0 | 35.71 |
| False Omission Rate (%) | 18.75 | 55.56 | 0 | 66.67 |
| False Positive Rate (%) | 16.13 | 42.86 | 0 | 71.43 |
| False Negative Rate (%) | 11.54 | 38.46 | 0 | 30.77 |

We found that interviewees who live at a distance between 74 and 154 m from Rua Miguel Petroni and avoid walking on the busiest street, consider the traffic speed a barrier to their displacements on foot, represented by node 12, a homogeneous node (Table A2). In addition, respondents who live at a distance of less than 74 m from the study road and know less than two neighbors on their side of the street, also belong to the class in which traffic speed is considered a barrier (Node 9).

On the other hand, some of the interviewees have another classification. Individuals who live between 74 and 154 m from Rua Miguel Petroni, who consider it the busiest street in the study area and do not avoid walking on it due to this, do not consider that traffic speed is an obstacle for pedestrians (Node 10). The details of each of the generated leaf nodes can be seen in Table A2.

The performance parameters for the Decision Tree and Random Forest models were generated from the confusion matrix. Through the Positive Predictive Value or Precision, it can be observed that a proportion of the sample of 90.20% and 72.73% (for the training and validation decision trees, respectively) and 100.00% and 64.29% (for the training and validation random forests, respectively) correctly identified the positive class, represented in this case by "0" (that is, correctly identified people who do not consider traffic speed a barrier). On the other hand, the Negative Predictive Value, which presented results of 81.25% and 100.00%, respectively, for the Training Decision Tree and Random Forest, reveals the proportion of correct identification of those who consider traffic speed a barrier (displayed by class "1"). The results of the other calculated parameters are shown in Table 2.

It is also observed that specificity (proportion of correctly identified negatives) evolves from 83.87% in the Decision Tree to 100.00% in the Random Forest, both for the training sample. However, for the validation sample, this value reduces from 57.14% in the solitary tree, to 28.57% in the set of trees that make up the Random Forest. This percentage shows the probability that the individual who considers traffic speed a barrier to has actually been classified as class "1". Sensitivity also improves the results of the tree in relation to the forest, showing, for example, that the true positive rate changes from 88.46% to 100.00% (training models) and from 61.54% to 69.23% (validation models) of the Decision Tree and Random Forest, respectively.

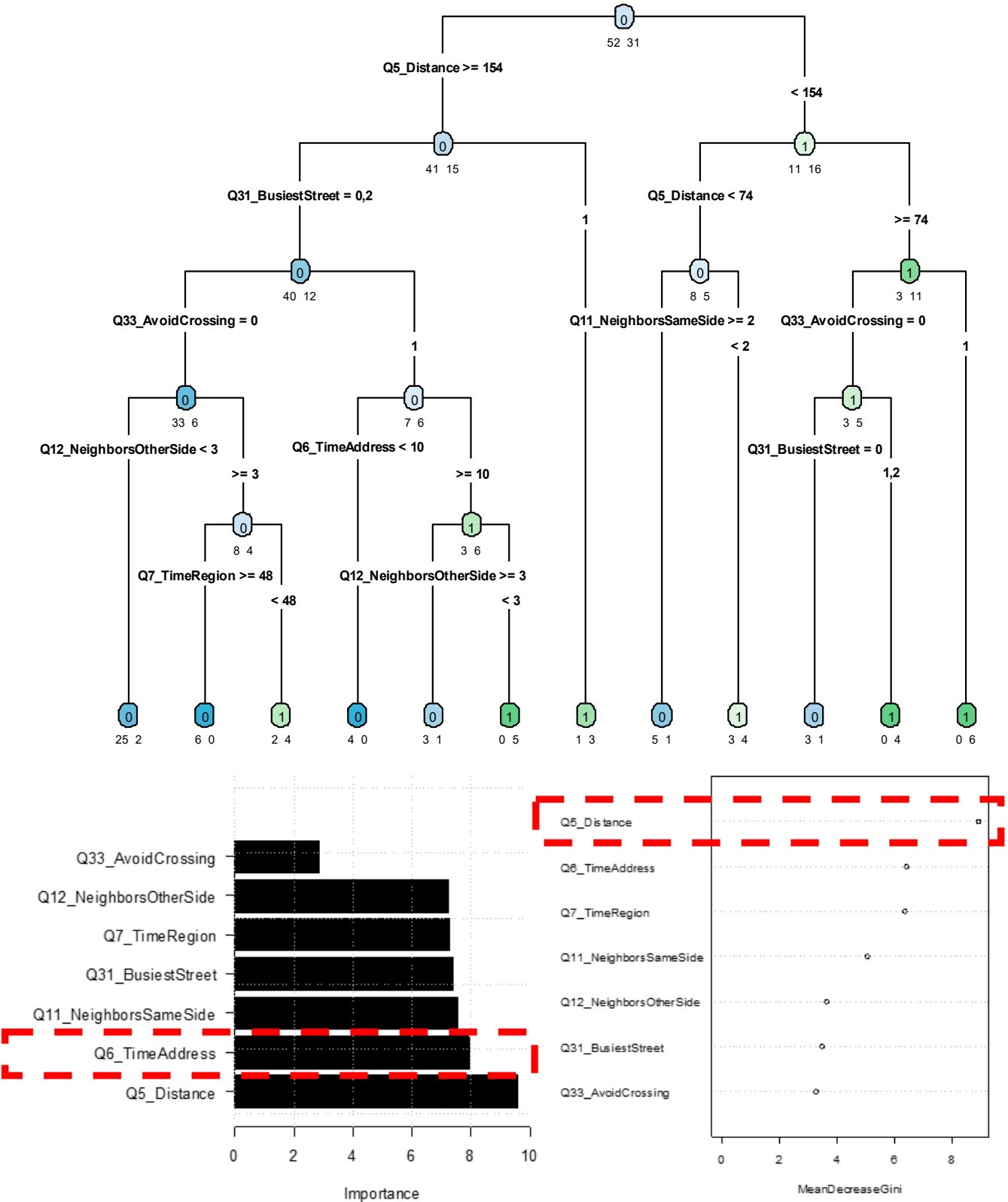

**Figure 2.** Decision Tree (Traffic Speed and Most Relevant Variables) for Training Database (n = 83).

### 4.2. Traffic Volume as a Dependent Variable

Regarding traffic volume, the questionnaire also showed the percentage of respondents who believe that this is an obstacle to their journeys on foot. In Class 1, accounting for 47.6%, are those who said that traffic volume is an obstacle to walking. Afterwards, the dependent

variable was analyzed using the Decision Tree and Random Forest techniques, aiming to classify the observations regarding traffic volume, identifying standard characteristics that lead the interviewees to choose each of the classes of the dependent variable and identifying the variables that most influence pedestrians' perception of it.

In the same way, from the variables that stood out as relevant in the analyses involving the traffic volume as a dependent variable and the characterization of the interviewees (i), the characteristics of social interactions (ii) and of travel and mobility (iii) as independent variables, the fourth analysis was made. In this case, the variables selected as independent were: group i—distance from the residence to Rua Miguel Petroni (Q5_Distance), time that the interviewee has lived in the study area (Q7_TimeRegion), time that the interviewee has lived at the current address (Q6_TimeAddress) and age group (Q2_Age); group ii—number of neighbors known on the same side and on the other side of the street he/she lives on (Q11_NeighborsSameSide and Q12_NeighborsOtherSide); and group iii—which is the busiest street close to where the person lives (Q31_BusiestStreet).

Continuing the process for the same configuration of dependent variable and independent variables, the Random Forest analysis was conducted. From the set of graphs of the importance of the variables, generated by both techniques, it can be seen that the distance from the interviewee's residence to Rua Miguel Petroni (Q5_Distance) appears as one of the variables that stands out as important for the prediction of both models. In addition, the age range of the interviewee (Q2_Age) and the time he/she has lived at the address and study area (Q6_TimeAddress and Q7_TimeRegion, respectively) also appears. The respective performance parameters are shown in Table 3.

**Table 3.** Predictive performance of AD and FA models for Analysis IV (Traffic Volume and the most relevant variables) (n = 83).

| Performance Measures | Decision Trees | | Random Forests | |
|---|---|---|---|---|
| | Training | Validation | Training | Validation |
| True positive | 42 | 6 | 44 | 6 |
| False positive | 13 | 6 | 0 | 6 |
| False negative | 1 | 4 | 0 | 4 |
| True Negative | 27 | 3 | 40 | 3 |
| Accuracy (%) | 83.33 | 47.37 | 100 | 47.37 |
| Sensitivity (%) | 97.67 | 60.00 | 100 | 60.00 |
| Specificity (%) | 67.50 | 33.33 | 100 | 33.33 |
| Prevalence (%) | 51.81 | 52.63 | 52.38 | 52.63 |
| Positive Predictive Value (%) | 76.36 | 50.00 | 100 | 50.00 |
| Negative Predictive Value (%) | 96.43 | 42.86 | 100 | 42.86 |
| False Discovery Rate (%) | 23.64 | 50.00 | 0 | 50.00 |
| False Omission Rate (%) | 3.57 | 57.14 | 0 | 57.14 |
| False Positive Rate (%) | 32.50 | 66.67 | 0 | 66.67 |
| False Negative Rate (%) | 2.33 | 40.00 | 0 | 40.00 |

The class considered positive for the dependent variable is "0".

Among the Decision Trees generated by alternating the composition criteria, the tree chosen for analysis (minsplit = 4, minbucket = 2, maxdepth = 5) was able to correctly predict 83.33% and 47.37% of the results for training and validation samples, respectively. The tree in question, made considering the traffic volume and the most relevant variables in relation to the interviewees' individual characteristics, social interactions and travel and mobility (Figure 3), classifies the sample into 11 leaf nodes.

We found that the interviewees who live at a distance of less than 191 m from Rua Miguel Petroni, consider it the busiest street in the study area, and are over 59 years old consider the traffic volume a barrier to walking (represented by Node 10, a homogeneous node). In addition, respondents who live at a distance of less than 191 m from the study road and consider another road as the busiest street in the study area also belong to the class that considers traffic volume a barrier, regardless of age (Node 11).

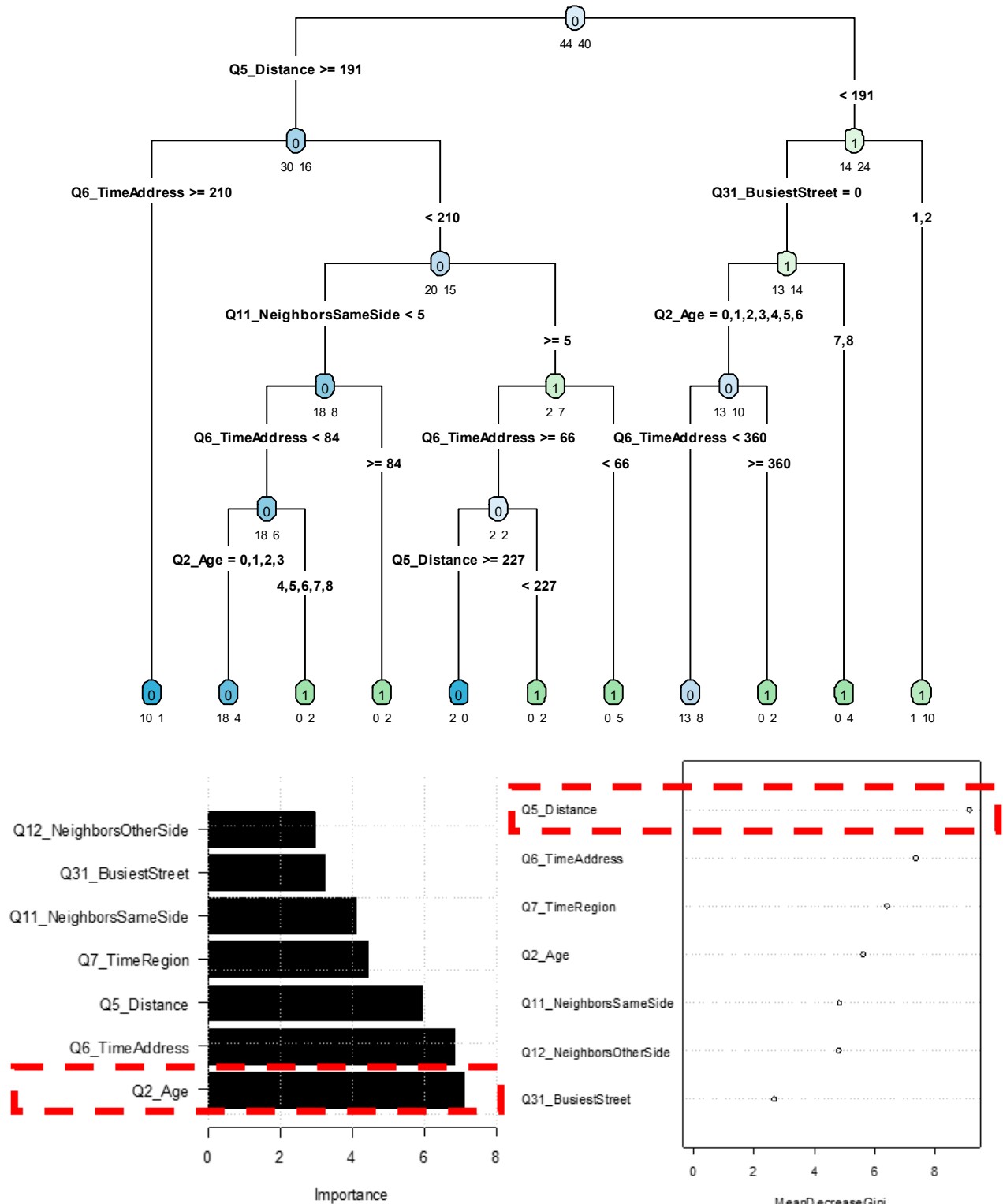

**Figure 3.** Decision Tree (traffic volume and the most relevant variables) for training database (n = 83).

On the other hand, some of the interviewees have another classification. Individuals who live at a distance of more than 191 m from Rua Miguel Petroni and have lived at the address for more than 210 months (17.5 years) do not consider that volume is an obstacle for pedestrians (Node 1).

In this case, the Positive Predictive Value, or Precision, shows that a proportion of the sample of 76.36% and 50% (for the training and validation trees, respectively) and 100% and

50% (for the training and validation forests, respectively) correctly identified the positive class, represented in this case by "0" (i.e., people who do not consider traffic volume a barrier). In addition, the Negative Predictive Value, which presented a result of 96.43% and 100%, respectively, for training Decision Tree and Random Forest modes, reveals the proportion of correct identification of those who consider traffic volume a barrier (shown by class "1"). A summary of the results is shown in Table 3.

It can also be observed that the Specificity evolves from 67.50% with the decision tree to 100% in the Random Forest, both for the training sample. However, for the validation sample, this value remains at 33.33% for the solitary decision tree and for the set of trees that make up the random forest. This percentage shows the probability that the individual who considers traffic volume a barrier has actually been classified as class "1". Sensitivity also presents an improvement in the result in the tree training sample in relation to the forest, showing, for example, that the true positive rate changes from 97.67% to 100%.

As seen, the application of the Decision Tree technique to the collected data classified the observations regarding the barrier effect, obtaining the independent variables that most influence the pedestrians' perception and identifying characteristics and patterns that seem to lead the interviewees to opt for each one of the dependent variable classes. At the same time, the Random Forest models contributed to defining the most important variables for the interviewees' perception of the barrier effect, allowing a comparison of the results with those obtained with the best performing Decision Tree.

## 5. Conclusions and Recommendations for Future Work

Previous research has reported that the barrier effect, resulting from a transport infrastructure and/or motorized vehicle traffic, affects not only the residents of a certain street, but also the residents and regulars of a surrounding area of influence. Some impacts recorded in the literature, arising from the barrier effect, concern the considerable change in the perception, behavior, travel, mobility and well-being of people who usually walk through the study area.

The present study sought, from the application of a comprehensive questionnaire, to collect enough information to identify whether there are indications that an important road in a medium-sized Brazilian city, selected for having high traffic of motor vehicles, acts as a barrier to the movement of pedestrians in the study area. This is the first research question (RQ1) formulated in the present study, which also evaluated the factors of greatest influence on pedestrians' perception of the barrier effect characterized by speed and traffic volume (RQ2) and looked for patterns between pedestrians' perception of the barrier effect and its characteristics (RQ3).

Residents, workers and/or regulars were interviewed in an area of up to 800 m from the study road. The set of 103 data collected was analyzed using descriptive statistics techniques and classification models. Within the classification models, we investigated the potential of Decision Trees and Random Forests to predict users' perception of the barrier effect, which was characterized in this study by two dependent variables: Speed and Volume (of motorized traffic).

Regarding the research question that aims to identify whether, in some respects, the selected road acts as a barrier to the study area (RQ1), it was observed that for 36.9% and 47.6% of respondents, traffic speed and traffic volume, respectively, represent a barrier to walking on the study road and on other roads in the study area. These results are similar to those found by Mindell et al. [2] and Anciaes [9].

Based on this finding, analyses were carried out in which traffic speed and volume figured individually as dependent variables, and questions pertaining to (i) individual characteristics, (ii) social interactions and (iii) travel and mobility as independent variables. Thus, we identified, in response to the second research question, that the most important variables for the pedestrian's perception of the barrier effect of the study area are: distance from the residence to the study route, time residing at the address and in the study area,

number of neighbors known on the same side and on the other side of the street of residence and the street considered the busiest street in the neighborhood.

Patterns were also identified among pedestrians' perception of the barrier effect in response to the third research question. It was observed, for example, that the distance that the pedestrian lives from the busiest street is the variable with the greatest influence on their perception of the barrier effect in the study area. In other words, the interviewees who live close to the study route pointed out traffic speed and traffic volume as barriers to their displacements on foot, as also mentioned by Mindell et al. [2], Powers et al. [36], and Lara [37].

It is important to mention that the questionnaire used, adapted from a study developed in England [24], proved to be adequate and efficient for collecting data on pedestrians' perception of the barrier effect also in the context considered. We believe that cities of other countries in the same population range of Brazilian medium-sized cities can use the proposed method. On the other hand, it is recognized that some changes can be made for future research. For example: (1) inserting a question about the distance that the interviewee considers walkable; (2) reconfiguring the age groups in order to better distinguish between respondents over 60 years of age; (3) considering those who claim not to travel on foot through the study area, which at first were disregarded, in order to identify the reason why this occurs; (4) modifying the approach involving the road median, as it can help in crossing the roads (instead of acting as a barrier); (5) inserting other dependent variables such as the number of lanes, percentage of heavy traffic, among others, as suggested by Anciaes and Jones [16].

This study is in line with at least one of the Sustainable Development Goals (SDG) proposed by the United Nations. In this case, SDG 11, which focuses on sustainable cities and communities and encourages actions to provide access to safe, affordable, accessible and sustainable transport systems for all, improving road safety, with special attention to the needs of those in vulnerable situations, such as women, children, persons with disabilities and older persons, in order to reduce inequality, among other targets. We hope this study can serve as a basis for decisions and public policies related to transport, in order to show the consequent impacts of motorized traffic and transport infrastructures on displacements by non-motorized modes and promote actions that prioritize and encourage walking and cycling. Some mitigation actions that can be carried out on roads where the barrier effect is observed are reducing the motorized traffic speed, implementing raised pedestrian crossings and traffic lights for pedestrians, reducing vehicle lanes, implementing bicycle lanes and/or cycle lanes, diverting heavy vehicle traffic to roads further away from the urban center, among others.

The methodology presented in this study contributes to future research in terms of identifying and characterizing the barrier effect and to the existing literature on the subject. The contributions are in the scope of data collection, the profile of the city and mainly with regard to the analysis through classification models (Decision Tree and Random Forest). The results obtained were relevant. However, some limitations are acknowledged. This is the case of the size of the sample, whose data collection in the field was interrupted due to the COVID-19 pandemic. It is believed that more robust results can be found in more representative samples from the study area. It is therefore suggested, for future work, to continue the collection and data analysis from the area in question, with larger and more representative samples. In addition, the proposed methodology could be applied to other medium-sized Brazilian municipalities, aiming to find patterns among the barrier effects existing in these cities.

**Author Contributions:** Conceptualization, A.N.R.d.S.; Data curation, M.C.R.d.J.; Formal analysis, M.C.R.d.J. and A.N.R.d.S.; Funding acquisition, Methodology, and Supervision, A.N.R.d.S.; Writing—original draft, M.C.R.d.J.; Writing—review & editing, A.N.R.d.S. All authors have read and agreed to the published version of the manuscript.

**Funding:** This research was funded by National Council for Scientific and Technological Development (308436/2015-6) and with the financial support from the Coordination for the Improvement of Higher Education Personnel (CAPES—Finance Code 88887.505770/2020-00).

**Institutional Review Board Statement:** For this type of study, which involves voluntary participation, ethical approval was required by the Ministry of Health—National Council of Health. So, the questionnaires through interviews were approved by the Research Ethics Committee for Human Beings at the Federal University of São Carlos, under Report No. 4,043,848, of 2020. In any case, the anonymity of all participants was assured; they were fully informed why the research was being conducted and that no personal risks were associated with the survey.

**Informed Consent Statement:** Informed consent was obtained from all subjects involved in the study.

**Data Availability Statement:** Not applicable.

**Conflicts of Interest:** The authors declare no conflict of interest.

## Appendix A

**Table A1.** Summary of the independent variables.

| Group of Independent Variables | Component Variables |
|---|---|
| (i) Characterization of the interviewees | Q1_Gender, Q2_Age, Q3_Education, Q4_PropertSituation, Q5_Distance, Q6_TimeAddress, Q7_TimeRegion, Q8 (two topics: car and motorcycly ownership), Q9_Income, Q10_Healthy. |
| (ii) Social interactions | Q11_NeighborsSameSide, Q12_NeighborsOtherSide, Q13_VisitNeighbors, Q14_ContactNeighbors and in Q15 some questions in topics about integration, vandalism, trust in people, fear of walking alone at night, cleanliness and other things about the study area. |
| (iii) Travel and mobility | Q16_DifficultyWalking, Q17_DifficultyMoving, Q18_PhysicalLimitation, Q19_DailyWalkingTrips, Q20_TravelTime, Q21 (some questions in topics about the preferences of the interviewee when walking), Q22 (some questions in topics about traffic conditions, crossings, traffic lights, pollution, relief, obstacles, road width, existence of median and other things about the study area), Q31_BusiestStreet, Q32_LiveOnBusiestStreet, Q33_AvoidCrossing; Q28, Q29 e Q30 about the traffic volume, traffic speed and time for crossing the street that he/she lives, respectively; Q34, Q35 e Q36 about the traffic volume, traffic speed and time for crossing the busiest street, respectively; and Q37_WalkingFrequencyOnRuaMiguelPetroni. |

**Table A2.** Details of decision tree leaf nodes (Traffic speed and most relevant variables).

| Leaf Node | Comments | % of Training Sample | Perception and Characteristics of Individuals | Situation |
|---|---|---|---|---|
| Node 1 | 27 | 32.5 | - Lives at a distance equal to or greater than 154 m from Rua Miguel Petroni.<br>- Considers Rua Miguel Petroni or another the busiest street in the study area.<br>- Does not avoid walking on the busiest street.<br>- Knows less than 3 neighbors on the same side of the street. | Speed is not a barrier |
| Node 2 | 6 | 7.2 | - Lives at a distance equal to or greater than 154 m from Rua Miguel Petroni.<br>- Considers Rua Miguel Petroni or another the busiest street in the study area.<br>- Does not avoid walking on the busiest street.<br>- Knows 3 or more neighbors on the same side of the street.<br>- Has lived in the study area for 4 years or more. | Speed is not a barrier |
| Node 3 | 6 | 7.2 | - Lives at a distance equal to or greater than 154 m from Rua Miguel Petroni.<br>- Considers Rua Miguel Petroni or another the busiest street in the study area.<br>- Does not avoid walking on the busiest street.<br>- Knows 3 or more neighbors on the same side of the street.<br>- Has lived in the study area for less than 4 years. | Speed is a barrier |
| Node 4 | 4 | 4.8 | - Lives at a distance equal to or greater than 154 m from Rua Miguel Petroni.<br>- Considers Rua Miguel Petroni or another the busiest street in the study area.<br>- Avoids walking on the busiest street.<br>- Has lived at the address for less than 10 months. | Speed is not a barrier |
| Node 5 | 4 | 4.8 | - Lives at a distance equal to or greater than 154 m from Rua Miguel Petroni.<br>- Considers Rua Miguel Petroni or another the busiest street in the study area.<br>- Avoids walking on the busiest street.<br>- Has lived at the address for 10 months or more.<br>- Knows 3 or more neighbors across the street. | Speed is not a barrier |

**Table A2.** *Cont.*

| Leaf Node | Comments | % of Training Sample | Perception and Characteristics of Individuals | Situation |
|---|---|---|---|---|
| Node 6 | 5 | 6.0 | - Lives at a distance equal to or greater than 154 m from Rua Miguel Petroni.<br>- Considers Rua Miguel Petroni or another the busiest street in the study area.<br>- Avoids walking on the busiest street.<br>- Has lived at the address for 10 months or more.<br>- Knows less than 3 neighbors across the street. | Speed is a barrier |
| Node 7 | 4 | 4.8 | - Lives at a distance equal to or greater than 154 m from Rua Miguel Petroni.<br>- Considers Rua Miguel João the busiest street in the study area. | Speed is a barrier |
| Node 8 | 6 | 7.2 | - Lives less than 154 m from Rua Miguel Petroni.<br>- Lives less than 74 m from Rua Miguel Petroni.<br>- Knows 2 or more neighbors on your side. | Speed is not a barrier |
| Node 9 | 7 | 8.4 | - Lives less than 154 m from Rua Miguel Petroni.<br>- Lives less than 74 m from Rua Miguel Petroni.<br>- Knows less than 2 neighbors on your side. | Speed is a barrier |
| Node 10 | 4 | 4.8 | - Lives less than 154 m from Rua Miguel Petroni.<br>- Lives at a distance equal to or greater than 74 m from Rua Miguel Petroni.<br>- Does not avoid walking on the busiest street.<br>- Considers Rua Miguel Petroni the busiest street in the study area. | Speed is not a barrier |
| Node 11 | 4 | 4.8 | - Lives less than 154 m from Rua Miguel Petroni.<br>- Lives at a distance equal to or greater than 74 m from Rua Miguel Petroni.<br>- Does not avoid walking on the busiest street.<br>- Considers Rua Miguel João or another the busiest street in the study area. | Speed is a barrier |
| Node 12 | 6 | 7.2 | - Lives less than 154 m from Rua Miguel Petroni.<br>- Lives at a distance equal to or greater than 74 m from Rua Miguel Petroni.<br>- Avoids walking on the busiest street. | Speed is a barrier |

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
