# Peer review of "Barrier Effect in a Medium-Sized Brazilian City: An Exploratory Analysis Using Decision Trees and Random Forests"

_sustainability, doi:10.3390/su14106309_

Round 1

Reviewer 1 Report

  1. Need to explain the sampling design - how are the samples selected and how was the sample size determined. Also, briefly explain the non-probabilistic sampling is applied in the selection of the respondents.
  2. A sample size of 103 was used, instead of the originally planned 385 samples. How does the reduced sample size affect the reliability and validity of the findings? Recall that you need 385 samples to achieve 95% confidence level with a sampling error of 0.05.
  3. How does staying length (>12 months) in current address significant in the study?
  4. In this study you define walking distance as 800m. However, 76.8% of your respondents live within 400m of Rua Miguel Petroni. Does it not make your findings biased towards those who live closer to Rua Miguel Petroni? How does this choice affect the reliability and validity of your findings?
  5. You reported a sample size of 103 (Line 246), but the findings (Table 2 and Table 3) were reported only from n=83 respondents. Please explain the discrepancy.
  6. In the conclusion, please explain the significance of this research/findings on pedestrian planning.

Author Response

Dear Editor,

Thank you for the opportunity to revise our paper. We also thank the reviewers for their careful reading of our previous draft and their thoughtful comments. We have taken their comments into consideration in preparing our revised manuscript. The following summarizes how we responded to the reviewers’ comments. The changes in the manuscript are likewise highlighted, but in green.

The following notes are aimed to clarify the doubts and changes required by “Reviewer #1” and incorporated in the new version of the paper entitled “Barrier effect in a medium-sized Brazilian city: an exploratory analysis using Decision Trees and Random Forests” submitted to the Sustainability journal, special issue Towards More Walkable and Liveable Cities: Perceptions, Attitudes, Methods, Technologies and Policies.

 C1: Need to explain the sampling design - how are the samples selected and how was the sample size determined. Also, briefly explain the non-probabilistic sampling is applied in the selection of the respondents.

The explanations about the sample are in page 6, as follows:

“At first, to ensure that the sample collected had statistical representation, considering a confidence level of 95% and a sample error equal to 0.05, we originally planned to collect data from 385 people. However, data collection was interrupted due to the Covid-19 pandemic and, in order to continue the studies, it was decided to increase the sampling error interval to 0.10, with the same level of confidence. In this case, the minimum sample size would be 96. The sample collected for this research, however, has a non-probabilistic profile, because, to reach the number of proposed questionnaires and due to limitations in the continuity of data collection, no strategy was adopted to select the homes and commercial establishments visited.” (Lines 256-264)

C2: A sample size of 103 was used, instead of the originally planned 385 samples. How does the reduced sample size affect the reliability and validity of the findings? Recall that you need 385 samples to achieve 95% confidence level with a sampling error of 0.05.

The highlighted part of this sentence, which was already presented to address the previous comment, was added to page 6 to address this issue.

“At first, to ensure that the sample collected had statistical representation, considering a confidence level of 95% and a sample error equal to 0.05, we originally planned to collect data from 385 people. However, data collection was interrupted due to the Covid-19 pandemic and, in order to continue the studies, it was decided to increase the sampling error interval to 0.10, with the same level of confidence. In this case, the minimum sample size would be 96. The sample collected for this research, however, has a non-probabilistic profile, because, to reach the number of proposed questionnaires and due to limitations in the continuity of data collection, no strategy was adopted to select the homes and commercial establishments visited.” (Lines 256-264)

C3: How does staying length (>12 months) in current address significant in the study?

This explanation was added to page 7, lines 273-277:

 “Some assumptions can be made about those interviewees who have lived a long time at their current address: they know the region better, as well as the positive and negative points; and have a more accurate perception of the region. On the other hand, they may have gotten used to the negative points in a way that they do not notice them with the right intensity.”

C4: In this study you define walking distance as 800m. However, 76.8% of your respondents live within 400m of Rua Miguel Petroni. Does it not make your findings biased towards those who live closer to Rua Miguel Petroni?

Not necessarily.

How does this choice affect the reliability and validity of your findings?

The highlighted part of this sentence was added to page 4, lines 151-154, to clarify this point:

“For this study, a distance of up to 800 meters was adopted, considered reasonable for walking in the region. Within this threshold we expected to find, in the region, people likely to walk on Rua Miguel Petroni. As a matter of fact, only people who knew and had already walked down that street answered the questionnaire.

C5: You reported a sample size of 103 (Line 246), but the findings (Table 2 and Table 3) were reported only from n=83 respondents. Please explain the discrepancy.

This difference is a consequence of the sample division in two groups: the first one for training and the second one for validation. The absolute numbers were now added to page 4, lines 176-177, to make this clear.

“To this end, it is assumed that the sample obtained is randomly divided into two groups, a part for training (80% of records - 83 interviewees) and a part for model validation (20% of records - 20 interviewees) and that the techniques are applied to four groups of different variables for each of the dependent variables.”

C6: In the conclusion, please explain the significance of this research/findings on pedestrian planning.

The following text was now added to the conclusions (Page 13, line 476-489):

“This study is in line with at least one of the Sustainable Development Goals (SDG) proposed by the United Nations. In this case, SDG 11, which focuses on sustainable cities and communities and encourages actions to provide access to safe, affordable, accessible and sustainable transport systems for all, improving road safety, with special attention to the needs of those in vulnerable situations, women, children, persons with disabilities and older persons, in order to reduce inequality, among other targets. We hope this study can serve as a basis for decisions and public policies related to transport, in order to show the consequent impacts of motorized traffic and transport infrastructures on displacements by non-motorized modes and promote actions that prioritize and encourage walking and cycling. Some mitigation actions that can be carried out on roads where the barrier effect is observed are: reducing the motorized traffic speed, implementing raised pedestrian crossings and traffic lights for pedestrians, reducing vehicle lanes, implementing bicycle lanes and/or cycle lanes, diverting heavy vehicle traffic to roads further away from the urban center, among others.”

Reviewer 2 Report

This paper analyzes the barrier effect of roads on walking trips by using Decision Tree and Random Forest classification models. Please see the detailed comments in the attached file.

Author Response

Dear Editor,

Thank you for the opportunity to revise our paper. We also thank the reviewers for their careful reading of our previous draft and their thoughtful comments. We have taken their comments into consideration in preparing our revised manuscript. The following summarizes how we responded to the reviewers’ comments. The changes in the manuscript are likewise highlighted, but in green.

The following notes are aimed to clarify the doubts and changes required by “Reviewer #2” and incorporated in the new version of the paper entitled “Barrier effect in a medium-sized Brazilian city: an exploratory analysis using Decision Trees and Random Forests” submitted to the Sustainability journal, special issue Towards More Walkable and Liveable Cities: Perceptions, Attitudes, Methods, Technologies and Policies.

C1: Statistical discription of the data, like sample size, major information and questions in the questionnaire is recommended to be presented.

We addressed this comment in the following sentences:

“At first, to ensure that the sample collected had statistical representation, considering a confidence level of 95% and a sample error equal to 0.05, we originally planned to collect data from 385 people. However, data collection was interrupted due to the Covid-19 pandemic and, in order to continue the studies, it was decided to increase the sampling error interval to 0.10, with the same level of confidence. In this case, the minimum sample size would be 96. The sample collected for this research, however, has a non-probabilistic profile, because, to reach the number of proposed questionnaires and due to limitations in the continuity of data collection, no strategy was adopted to select the homes and commercial establishments visited.” (Page 6, lines 256-264)

“The questionnaire has 37 questions. It was designed to allow various analyses, involves questions related to the characterization of the interviewee (age, academic background, income, address, health status and other things); social interactions (for example: contact with neighbors, integration into the region, trustworthiness of people, fear of walking alone at night, cleanliness of the region); travel and mobility (walking difficulties, disabilities, number of walking trips and duration thereof, existence of a crosswalk, whether the speed and volume of traffic represent barriers to reaching a destination on foot, characterization of the street where the person lives, the region or street that he/she considers the busiest and other aspects); and, finally, an open space for the respondent to express his/her opinion (in this part, the answers were audio recorded). A summary of the independent variables considered is presented in Appendix A.” (Page 4, lines 130-141)

C2: It is recommended that the authors give the summary of independent variables and the corresponding serial number in the questionare.

We added this information to Appendix A.

C3: What is the defination of medium-sized city? What are Brazilian standards?

We added the following sentence to Page 4 (Lines 127-128) to answer the questions above:

 “The methodology was applied to a medium-sized city, which has a population between 100 and 500 thousand inhabitants, by Brazilian standards [23].”

C4: How can we transfer the findings in this paper to cities in other country?

We added the following sentence to Page 13 (Lines 467-468) to answer the question above:

“We believe that cities of other countries in the same population range of Brazilian medium-sized cities can use the proposed method.”

C5: As shown in Fig. 1, how to determine the boundry of the are?

The boundary of the area was determined as an 800-meter buffer around the street of interest, which is easily reached by foot. We added the following sentence to Page 6 (Lines 236-239) to make this clear:

“That part of the street was selected, similarly to what was done in other studies [2; 5; 6; 9], because of previous reports of the population describing difficulties for crossing and dangerous situations for displacements by active modes.”

C6: Why it is called dependent variable? Is it the influencing factor that a walker decide that a road is a barrier or not? If it is true, maybe it should be called independent variable.

Traffic volume and speed are treated as dependent variables because they are believed to be responsible and characterize variables for the barrier effect. The independent variables are those that influence (or not), the interviewee's perception of the barrier effect.

Reviewer 3 Report

Dear Authors

Initially, I greet them in the hope of finding them in good health. Next, I congratulate you for the opportunity you have given me to read an article of this quality. After reading manuscript 1710630, I report that the authors were innovative in studying the predictive potential of Decision Tree and Random Forest models to classify respondents' perception of the effects of barriers on urban roads. I consider it a work that sought to integrate theory with people's real needs. Regarding the literature review, I believe it is adequate and current for the case in question. Regarding the proposed methodology, it was described in a clear and objective way. As for the results, they were precisely exposed. I believe that this research work needs some minor revisions.

specific comments:

1.Despite the explanation of the choice of the research site, contained in lines 219-229, it would be interesting to indicate another place to serve as a comparison. Have the authors identified similar research in the literature? Would you like me to explain this question further? 2. In lines 131-133 the researchers inform the means of data collection. on the lines. in 141-143, the authors inform the times and days of collection or approach of the interviewees. On line 239, the authors inform that the collection was interrupted due to COVID-19. The authors did not specify the period of beginning and end of the collection of information, including this information. And explain better why the approach of users at the bus stop? Why didn't you place approach points inside the selected area?

Yours sincerely
reviewer

Author Response

Dear Editor,

Thank you for the opportunity to revise our paper. We also thank the reviewers for their careful reading of our previous draft and their thoughtful comments. We have taken their comments into consideration in preparing our revised manuscript. The following summarizes how we responded to the reviewers’ comments. The changes in the manuscript are likewise highlighted, but in green.

The following notes are aimed to clarify the doubts and changes required by “Reviewer #3” and incorporated in the new version of the paper entitled “Barrier effect in a medium-sized Brazilian city: an exploratory analysis using Decision Trees and Random Forests” submitted to the Sustainability journal, special issue Towards More Walkable and Liveable Cities: Perceptions, Attitudes, Methods, Technologies and Policies.

C1: Despite the explanation of the choice of the research site, contained in lines 219-229, it would be interesting to indicate another place to serve as a comparison. Have the authors identified similar research in the literature? Would you like me to explain this question further?

We added a sentence to address the point raised by the reviewer:

“The specific area of the municipality selected for the study comprises a stretch of about 1.5 km of Rua Miguel Petroni (Figure 1 - in Portuguese, Rua means Street), an urban road with intense traffic of motor vehicles and 4.9 km in total length. That part of the street was selected, similarly to what was done in other studies [2; 5; 6; 9], because of previous reports of the population describing difficulties for crossing and dangerous situations for displacements by active modes.” (Page 6, lines 236-239)

C2: In lines 131-133 the researchers inform the means of data collection. on the lines. in 141-143, the authors inform the times and days of collection or approach of the interviewees. On line 239, the authors inform that the collection was interrupted due to COVID-19. The authors did not specify the period of beginning and end of the collection of information, including this information.

This information was now added to Page 7, Line 266.

“A total of 103 respondents were interviewed, in March 2020, of which 66% were female, predominantly aged between 30 and 59 years, most of whom had completed high school, and about 86.4% considered their health status to be excellent or good.”

C3: And explain better why the approach of users at the bus stop? Why didn't you place approach points inside the selected area?

We did not place the approach points inside the selected area because we did not consider this information relevant for the study, given that some individuals were not interviewed in their residences and some were. In any case, we looked for frequent walkers in the region and, more specifically, on the street of interest. 

Round 2

Reviewer 1 Report

Thank you for addressing my concerns. The revised version has increased the quality of the paper significantly.

Author Response

all  the revision has been done
